# A Systematic Review of Traumatic Brain Injury in Modern Rodent Models: Current Status and Future Prospects

**DOI:** 10.3390/biology13100813

**Published:** 2024-10-11

**Authors:** Evgenii Balakin, Ksenia Yurku, Tatiana Fomina, Tatiana Butkova, Valeriya Nakhod, Alexander Izotov, Anna Kaysheva, Vasiliy Pustovoyt

**Affiliations:** 1Federal Medical Biophysical Center of Federal Medical Biological Agency, 123098 Moscow, Russia; 2Institute of Biomedical Chemistry, 119121 Moscow, Russia

**Keywords:** preclinical TBI models, traumatic brain injury, animal TBI models, FPI model, CCI model, WDI model, DAI model, BW model, CHIMERA model, ESW model

## Abstract

**Simple Summary:**

Traumatic brain injury is a leading cause of death and disability worldwide, according to the Centers for Disease Control and Prevention. It results in various long-term consequences, including brain atrophy, nerve damage, and significant economic costs. Traumatic brain injury is most common among working-age adults, highlighting the urgent need for improved diagnostic and treatment methods. Animal models serve as a valuable tool for studying the mechanisms of traumatic brain injury and developing new treatments. These models allow researchers to control injury parameters and analyze post-traumatic effects, advancing our understanding of the condition.

**Abstract:**

According to the Centers for Disease Control and Prevention (CDC), the national public health agency of the United States, traumatic brain injury is among the leading causes of mortality and disability worldwide. The consequences of TBI include diffuse brain atrophy, local post-traumatic atrophy, arachnoiditis, pachymeningitis, meningocerebral cicatrices, cranial nerve lesions, and cranial defects. In 2019, the economic cost of injuries in the USA alone was USD 4.2 trillion, which included USD 327 billion for medical care, USD 69 billion for work loss, and USD 3.8 trillion for the value of statistical life and quality of life losses. More than half of this cost (USD 2.4 trillion) was among working-age adults (25–64 years old). Currently, the development of new diagnostic approaches and the improvement of treatment techniques require further experimental studies focused on modeling TBI of varying severity.

## 1. Introduction

In modern society, injury is one of the topical problems, both medical and social, related to its abundance among young able-bodied people, high mortality rate, severity of disability, and significant economic costs for the treatment and rehabilitation of patients [1,2]. Traumatic brain injury (TBI) is a leading traumatic injury that is characterized by a mortality rate of 35–38% and a disability rate of about 50% [3,4]. Globally, annual TBI statistics amount to 1.5 million deaths and 2.4 million disabilities. In the United States, the annual incidence rate of TBI is approximately 1.6 million, including 51,000 deaths and 124,000 long-term disabilities [1]. According to the Centers for Disease Control and Prevention (CDC), the national public health agency of the United States, traumatic brain injury is among the leading causes of mortality and disability worldwide [5,6]. The causes of TBI include social, demographic, geographic, climatic, and other factors. For example, the leading causes are road traffic accidents in the United States, motor scooter injuries in Taiwan, and falls from height in Scotland [7,8]. Population-based epidemiological studies conducted in various countries at the end of the 20th century played a significant role in the investigation of TBI causes and prevalence. The rate of TBI per 1000 people varies significantly: 7.3% in China, 5.3% in the USA, 9% in Russia, and 1.1% in Scotland. Injuries in males are 2- to 3-fold higher than those in females, with this trend being similar in all age groups, except infants and the elderly. TBI is most common in males between 20 and 39 years of age. Mild TBI predominates in clinical presentations (80–90%) [9].

TBI can have a variety of sequelae that include diffuse brain atrophy, local post-traumatic atrophy, arachnoiditis, pachymeningitis, meningocerebral cicatrices, cranial nerve lesions, and cranial defects [10].

The development of new approaches to the diagnosis and treatment of TBI patients has used primarily animal models that have a number of advantages: a limited range of injury mechanisms (animal models enable control of impact conditions, which provides a relatively uniform injury mechanism), genetic and demographic homogeneity (animal models usually have a genetically homogeneous population, which reduces variability in results due to genetic differences), and controllable injury parameters (injury severity can be predetermined, which enables studies using models that replicate different levels of TBI severity). Animal research is considered a key element in the advancement of biomedical science. Although its use is controversial and raises ethical challenges, the contribution of animal models in medicine is essential for understanding the physiopathology and novel treatment alternatives for several animal and human diseases [11]. The described advantages of biological test systems make them a valuable tool for studying TBI mechanisms and consequences and for evaluating new treatments [12,13].

## 2. Materials and Methods

### 2.1. Search Criteria

To conduct this review, we carried out a comprehensive literature search in the PubMed database to identify relevant publications.

Inclusion Criteria: We applied several inclusion criteria to select studies. Only full-text articles in English that described TBI models in animals, particularly mice, were considered.

Exclusion Criteria: The exclusion criteria included articles published in languages other than English, general reviews, articles with insufficient data, articles without full-text access, and studies involving other types of injuries without a detailed investigation of TBI.

All identified articles were analyzed and screened. The evaluated articles were published between May 2014 and May 2024. A total of 4948 articles were identified using the search criteria. Subsequently, 3967 articles were excluded due to a lack of relevant data (such as full-text access, publication date within the last 10 years, use of species other than mice, classical articles, clinical studies, clinical trials, meta-analyses, reviews, or systematic reviews). The remaining 133 articles were further assessed for eligibility and selected for inclusion in the study. A flowchart of the search strategy is shown in Figure 1.

### 2.2. Inclusion/Exclusion Criteria

Abstracts and titles were selected exclusively from peer-reviewed primary research reports specific to animal models of traumatic brain injury (TBI) (FPI, CCI, WDI, DAI, BW, CHIMERA, and ESW models). All other articles were excluded.

### 2.3. Retrieval of Information from Full-Text Articles

To systematically retrieve data from the selected articles, we developed a structured data collection form that was implemented using the Google Forms platform. This form collected general study identifiers, such as the title, first and last authors, date of publication, and name and description of the TBI model used.

Specific information related to TBI induction techniques included identification of the injury device, anesthesia use, surgery indicator, device type, head fixation method, animal stabilization method, geometric characteristics of the impact tip (size, shape, and material), impact location and surface, weight drop parameters (weight, height), tube composition, and additional damping materials.

## 3. Results

The sample of 133 articles was divided into three main categories of injury models. The largest group was the blast wave impact model (*n* = 39), followed by the fluid percussion injury (FPI) model (*n* = 20), and then all “other” models (*n* = 74). The sample included the following models: the FPI model (*n* = 20), the controlled cortical impact (CCI) model (*n* = 14), the weight drop injury (WDI) model (*n* = 15), the diffuse axonal injury (DAI) model (*n* = 6), the blast wave (BW) model (*n* = 39), the CHIMERA model (*n* = 19), the explosive shock wave (ESW) model (*n* = 11), and models that use neurospecific markers indicating blood–brain barrier disruption (*n* = 9).

### 3.1. TBI Pathogenesis

The mechanism of (TBI) is a complex process that results from a combination of primary and secondary impact mechanisms and leads to temporary or permanent impairment of neurological function [14]. Primary injury is directly related to direct physical impact to the brain [15,16,17]. Secondary changes can occur within minutes or days after the primary impact and include molecular, chemical, and inflammatory responses associated with the progression of brain injury. These responses involve neuronal depolarization followed by the release of excitatory neurotransmitters, such as glutamate and aspartate, which induce an intracellular calcium surge. An elevated intracellular calcium level activates various enzymes, such as caspases and calpain, and free radicals, which leads to cell destruction through apoptosis. This process of neuronal degradation is accompanied by an inflammatory response that increases neuronal damage and disruption of the blood–brain barrier (BBB), leading to cerebral edema [13,17,18,19,20,21,22]. The secondary damage phase is followed by a recovery period that includes anatomical, molecular, and functional reorganization [23,24,25,26,27,28,29].

The Monro–Kellie doctrine has been a well-established principle of intracranial hemodynamics for over 200 years. Its fundamental concept is simple: the combined volume of brain tissue, blood, and cerebrospinal fluid (CSF) within the skull remains constant [30]. The intracranial space is filled with the brain (83%), cerebrospinal fluid (11%), and blood (6%). The relationship between these components provides a homeostatic intracranial environment. However, an increase in the volume of any of the intracranial space components activates a cascade of compensatory mechanisms. An increase in the volume of intracranial space contents often results from massive blood influx and impact, causing cytotoxic and vasogenic edema as well as venous stasis. The brain is incompressible, so edematous brain tissue initially leads to the redistribution of some cerebrospinal fluid into the spinal canal. Over time, blood, especially that of venous origin, is also removed from the brain due to compensatory mechanisms. In the absence of timely intervention, even the maximum compensatory mechanisms fail, which leads to pathological brain compression and death.

### 3.2. Basic Requirements for an Experimental Model

Currently, experimental TBI models require the induction of injuries similar to those seen in humans. Regardless of the physiological, behavioral, or anatomical parameters used to assess the response to injury, it is important that the results are reproducible, quantifiable, and generated under controlled injury severity. No one model can accurately replicate all the features of mechanisms of human TBI. However, this review discusses several preclinical models for a proper description of the underlying pathology.

Any experimental TBI model should satisfy certain criteria: conformity of age, sex, body weight, and genetic characteristics of used animals, housing conditions, and circadian rhythms to the human’s pathology as well as no differences in these parameters between the control and study groups. This definitely helps to get more precise results with high correlation regarding people’s disease. It is also important to clearly define the physical parameters, including the exact location and specific degree of injury. The central nervous system response to injury should be measurable, quantifiable, and replicable across different laboratories. The degree of injury should be consistent with the mechanical force applied to the head or brain and characterized by a specific dependence.

Modern TBI models include standardized experimental protocols and surgical techniques. An important aspect is control measurements in special sham-operated animals that undergo the same surgical procedures as animals in the experimental group, except the injury, e.g., access to the brain for injury equipment, anesthesia, maintenance of body and brain temperature, introduction of intracranial probes and cannulas, etc. [31,32]. The mechanical characteristics underlying the degree of injury are evaluated using computer measurements of the applied load that, depending on the model, may include pressure gradients of a fluid acting on the brain, the impactor velocity, and the energy of forces causing head acceleration or deceleration. This enables the tuning of injury devices and maintaining a narrow range of changes in injury severity within a single experiment.

### 3.3. TBI Models

The perfect modeling of all aspects of TBI within a single test system has proven impossible, so the animals used in TBI models are selected with allowance for specific features. For example, pigs provide a valuable biological TBI model because their physiology and size are similar to those of humans, which enables important physiological monitoring. In addition, the pig brain has a complex cellular structure close to that of the human brain, in contrast to that of small rodents [31,32].

Although large animals may more closely replicate the mechanical aspects of human TBI, and some models, such as primates, may better reflect the neurophysiological processes and functional impairments seen in humans, rodent models remain the most common ones in TBI research [33]. The advantages of rodent models include their relatively low cost, small size, reduced impact on regulatory systems, availability of genetically modified lines, and low variability of measurements. In addition, the relatively short lifespan of 2 to 3 years in most mouse and rat strains offers a significant advantage for aging research, a topic that will be discussed in more detail below [13].

To date, several different TBI models have been developed and used experimentally (Table 1), each of which has its own advantages and disadvantages [13,23,34,35,36,37,38,39,40,41,42,43,44].

### 3.4. Fluid Percussion Brain Injury (FPI) Model

The fluid percussion injury model can induce diffuse (central FPI) and mixed (focal and diffuse; lateral FPI) pathological changes that are frequently seen in humans after TBI [59,60,61]. This is the most commonly used and studied rodent TBI model. FPI has been successfully used in various test systems, including rabbits, cats, rats, mice, and pigs [37,39,40,44]. Injury (after scalp incision and trepanation) results from the rapid administration of a bolus of fluid to the intact dura mater, followed by the concentric spreading of the fluid into the epidural space, which causes a diffuse effect on the brain [36]. In this model, the injury can be medial (the burr hole is made in the midline) or lateral. Fluid pressure can be adjusted to simulate mild, moderate, and severe TBI [36,62,63]. This model replicates both local traumatic changes (injured cerebral cortex areas with petechial, intraparenchymal, and subarachnoid hemorrhages) and traumatic changes distant to the primary source (in the hippocampus, thalamus optica, and others), which are caused by secondary brain damages [64,65]. The fluid percussion TBI model enables assessing motor and cognitive impairments caused by injury and their changes during treatment [45,47,66].

The disadvantage of this method is that the use of this model, especially the central FPI model, is associated with highly severe injuries and mortality due to uncontrolled brainstem injury and neurogenic pulmonary edema. In terms of biomechanics, this model is most distant from human TBI, and researchers suggest that axonal damage in distant structures, such as the hippocampus and thalamus, results from secondary neurochemical reactions [13,46,48]. This model lacks standards characterizing injury parameters, such as differences in peak pressure and impact duration, which significantly affects the reproducibility and comparability of research results. In the FPI model, responses to injury can vary significantly even for the same impact parameters, which complicates data interpretation. The craniectomy procedure required in the FPI model may lead to complications, such as infection and hemorrhage [67,68]. These limitations should be considered upon interpretation of the results of research based on the FPI model and their extrapolation to clinical practice.

### 3.5. Controlled Cortical Injury Model

This model uses the impact of a solid impactor on the intact dura mater of the brain [50,69]. In this case, the animal’s head is usually fixed. The impactor is driven by a pneumatic [70,71,72] or electromagnetic [73] device that enables controlling the time, rate, and depth of impact to the brain [74,75]. The use of low mechanical energy enables the simulation of concussion, but this model is mainly used to study focal injuries, in particular TBI, accompanied by epidural and/or subdural hematomas [76]. Like the fluid percussion model, this model also allows studying traumatic changes in distant, injury-sensitive brain areas, such as the hippocampus, dentate gyrus, and optic tubercle, and assessing movement disorders, changes in motor coordination, and cognitive deficits [77,78,79].

This and previous methods cause severe brain injury, neuroinflammation, and behavioral impairments, including cognitive impairments. However, they have similar disadvantages, such as the rapid spontaneous recovery of brain function (within 2 weeks) and technical difficulties in modeling injuries [49,51,80].

### 3.6. Weight Drop Injury (WDI) Model of TBI

There are two ways to parametrically modulate the intensity of impact injury: changing the weight and initial height of the object [81]. The animal’s head is fixed, but researchers have not ensured this in most studies. After incising the scalp and exposing the surface of the skull, the injury is produced under general anesthesia. The severity of TBI is varied by adjusting the weight and height of the load [53,54,58]. In 1994, A. Marmarou proposed focal trauma that is simulated using a brass weight dropped from a predetermined height through a plexiglass tube. The weight of the object varies from 20 to 200 g, and the drop height can reach 2 m. Skull fracture is prevented by resting the rat’s head on a foam pad instead of a hard plastic disk. A metal helmet protects the exposed skull and serves as a target for the weight [57]. The short time required to prepare the animal for injury and lack of need for a surgical skull hole make this model simple and convenient [56,62].

The disadvantage is the high rate of skull fractures upon modeling severe TBI [58]. Like other rodent TBI methods, this model reveals a variety of brain injuries, ranging from mild brain injuries emulating concussions to focal contusions (where impact is applied to the skull) that are accompanied by long-term neuronal loss. Impairment of motor and cognitive functions was also observed in this model [13,46,48,55,62,66].

To overcome the risk of skull fractures that might not occur in human TBI, the model was modified [58,82]. After incising the scalp and exposing the rat’s skull, a protective “helmet”—a round steel plate of 1 cm in diameter—is firmly fixed to the bone (using dental cement). The injury is induced by dropping a weight with a blunt surface onto the “helmet” from a height, which ensures head acceleration upon minimal local impact at the point of application of the traumatic force. The animal’s head is not fixed. The “helmet” ensures the wide spread of the traumatic force across the skull surface. As in the previous method, the severity of injury is regulated by the load weight and drop height. This model is characterized mainly by diffuse brain injury, local injury of the cortex adjacent to the injured skull area, and cell death in the injury-sensitive brain area (hippocampus) [83,84].

### 3.7. Diffuse Axonal Injury (DAI) Model of TBI

The modeling of TBI in animals uses ether anesthesia and fixation of the limbs in the supine position [85]. The surgical level of anesthesia is assessed based on the lack of a corneal reflex on the right side of the animal’s skull. The hair on the parietal region is cut off, and the site is treated with an aseptic solution. A longitudinal skin incision is made, and the skull bones are trepanned at a distance of 2 mm from the midline using a cranial bone bur, while keeping the dura mater intact. The weight is a 114.6 g steel cylinder that is dropped from a height of 20 cm along a guide polyethylene tube, providing maximum impact to the trepanation window area in the right parietal region of the brain, with an impact force of 0.224 N. After injury, the animal’s wound is sutured with surgical thread (0.2 mm); the sutures are treated with an antiseptic, and a gentamicin solution is administered intramuscularly [75,86,87,88].

Modeling TBI in animals using ether anesthesia promotes the development of persistent neurological deficit, with the animal remaining alive; spontaneous recovery of brain function after 4–6 weeks; and control and evaluation of the therapy efficacy, in particular cell therapy, to restore brain function after TBI. A technical result using ether anesthesia is achieved by the following: influencing the cortical motor areas in the fronto-parieto-temporal region and preventing rupture of the dura mater and deep damage to brain tissue (2 mm), provided that the impact energy does not exceed 0.224 N. It should be taken into account that injuries caused by an impact energy of more than 0.224 N can lead to the death of the animal or result in severe focal lesions that lead to neuronal destruction, glial scar formation, and persistent neurological deficit. Conversely, injuries caused by an impact energy of less than 0.224 N almost do not induce severe motor neurological deficits and can sometimes cause mild hemiparesis/monoparesis that resolves within 5 to 7 days [89].

### 3.8. Blast Wave (BW) Model of TBI

Blast-related traumatic brain injuries are one of the most common injuries sustained by soldiers and veterans serving on military installations, which, in turn, has necessitated the development of a mechanistically relevant TBI model [90,91,92,93]. Over the years, various injury techniques have been developed, e.g., shock tubes [94,95,96], open space explosions [23,38,97], and blast tubes to simulate blast-induced injuries [42,98] in humans. Of these, the most widely popular laboratory models [23,38,42,99] are those where blast waves (shock wave plus gust of wind) are generated through the detonation of an explosive charge [13,49,51,52,80]. Currently, there are no standardized parameters for shock tubes (e.g., type of gas or explosive, tube design), test system type and location, protection of the body or head mobility, and peak overpressure or its duration. These factors can significantly affect the nature of injury, emphasizing the need for a critical approach to studies that use gaseous media (e.g., air, nitrogen, helium). Differences in the implementation of blast research regulations partly explain the variability in threshold values and pathological changes reported by laboratories from different countries. Given the recent development of various models and the lack of clinical and neuropathological descriptions of blast-induced TBI, these models are perhaps the most diverse experimental TBI models.

When blast tubes are used in an animal TBI model, head immobilization with a metallic holder is necessary to minimize the inertial forces acting on the skull [41]. The shock tube model, which is based on the compression–expansion mechanism of chambers separated by a membrane, can generate high-speed shock waves with controlled peak pressure due to the rupture of a membrane of a certain thickness [100]. The closed-head shock tube model replicates the brain injury mechanism through rapid angular acceleration, which determines the magnitude of the mechanical load based on the spatial location of the animal in the device [98,101,102,103]. This model is used in experiments on various animal species (rodents, pigs, primates), leading to focal and diffuse neuronal injury as well as edema and ischemia typical of TBI [42,97].

Shock wave tubes can vary significantly in size, from centimeters to tens of meters in length. A membrane, e.g., a mylar membrane, separates the driver and driven section of the tube. Compressed air or other gas fills the driver section to a pressure sufficient to rupture the membrane. This produces a shock wave that propagates through the driven section. Experimental samples (animals or materials) are located inside or outside the driven section at a different distance to manage the environment conditions and severity of damage (Table 2).

Disadvantages of the shock tube include the differences in the brain size, geometry, and white/gray matter ratio, which complicates the comparison of animal models with humans. The mass effect induced by injury differs between large human brains and smaller rodent brains, which requires scaling of injuries to get a similar effect. The shock tube method used for modeling head injuries is not standardized, which leads to variability in results and may cause additional injuries, such as hypoxia and blood vessel damage [66,104,105,106].

The pathophysiological features of brain injury induced by low-intensity blast exposure, given the abundance and nature of TBI in combat settings, require an urgent need to study its underlying pathological mechanisms. In 2018, H. Song et al. proposed a model of low-intensity blast (LIB) brain injury that replicates the mechanical impact to the body with parameters of 46.6 kPa and a maximum impulse of 8.7 psi (or 60 kPa/ms), which is significantly different from a high-intensity blast (>100 kPa) [114,115]. Despite the absence of lethal or gross macroscopic damages, this model leads to measurable impairments in neurobehavioral functions, decreased mitochondrial fission–fusion activity, bioenergetic deficiency, increased oxidative stress, reduced mitophagy, increased compensatory respiration, and axonal myelin sheath degradation [108]. Importantly, damage to myelinated axons is most pronounced in the subacute period (7 days after injury), in contrast to the chronic period (30 days after injury). A single LIB exposure causes an increase in total tau, phosphorylated tau, and amyloid-β [116].

The disadvantage of the blast injury method is associated with a deep understanding of blast physics. A model that is aimed at replicating the multifaceted effects of blast injury needs to accurately define the parameters typical of combat settings. The lack of precise specification reduces practical value and leads to the risk of misleading conclusions. The widespread use of disparate experimental conditions in different models complicates the comparison and generalization of the research results [108,109,110,117].

Although each injury model has its own characteristics, none of the models can entirely replicate the full range of pathologies related to human TBI. Of the three common experimental animal TBI models, the fluid percussion, controlled cortical injury, and weight drop/acceleration impact ones are the most important. The fluid percussion device produces injury by a brief pulse of fluid pressure to the dura mater through a craniectomy. Unlike the fluid percussion model, controlled cortical injury causes mechanical damage to the dura mater through a directed forceful blow by using pneumatic pressure. The weight drop/weight impact model involves dropping a rod of a specific mass onto a closed skull. The TBI models listed above require the use of anesthetics at the time of injury for ethical reasons. Several anesthetics, including isofluorane and ketamine, are known to have neuroprotective effects, improving functional and histological outcomes in TBI models when present at the time of injury [69,118,119]. This limits clinical application because patients have no anesthetic agents at the time of injury.

### 3.9. Closed-Head Impact Model of Engineered Rotational Acceleration (CHIMERA) Model

TBI associated with physical activity and traffic accidents is usually caused by the sudden acceleration of the head, which causes abnormal changes in cerebral tissue that are different from the effects of direct localized impact to the head [120,121]. This type of injury affects key white matter regions whose damage is thought to result from the shear forces typical of rotational acceleration. These injuries induce inflammatory processes that can persist long after the initial traumatic event.

Given the abundance of TBI, a novel model has been developed to rapidly and non-invasively induce injury by closed-head impact using engineered rotational acceleration (CHIMERA). Recent studies using the CHIMERA model have revealed not only the underlying pathological post-traumatic changes but also concomitant subtle changes in the accumulation of misfolded proteins, receptor expression, and neuronal death [120,122,123]. CHIMERA differs from existing neurotrauma simulation systems in that it uses a completely non-surgical procedure to precisely impact a closed skull with defined dynamic characteristics of traumatic impact, which enables kinematic analysis of unrestricted head movements [124,125].

Researchers have achieved significant progress in applications of the model by focusing on what is needed for its testing. However, most CHIMERA studies have used only adult male mice [126,127]. The development of this model requires further work with female animals of different age groups as well as research aimed at generating and standardizing additional models for methodological studies of clinical relevance.

Experimental rodent TBI models have limited validity because they do not fully account for the biomechanical and physiological features of human injury. In particular, rodent brains are largely lissencephalic (no gyri), while human brains are gyrencephalic (with gyri). These differences in anatomy may significantly influence the dynamic characteristics of intracranial brain movement in TBI, leading to varying degrees of deformity and injury [32].

In non-military settings, a significant number of head injuries are reported in motor vehicle accidents, which do not involve a direct impact to the head but result from the rotational forces of the collision. These traumas lead to diffuse axonal injury to the brain [128,129]. Currently, there are no adequate rodent models to replicate this type of injury. The studies on pigs were the only models to come close to imitating the diffuse axonal brain injury associated with rotational forces [109,130,131,132,133].

The use of large animals for TBI research has several limitations. First, large animals require significant financial expenses on both acquisition and maintenance, as well as specialized housing. Second, the long lifespan of large animals implies increased requirements for care both before and after injury. However, the use of large animals in TBI research has unique advantages. Large animal models more accurately replicate the cortical injury mechanisms typical of clinical settings and provide the opportunity to comprehensively investigate neurorecovery after TBI [134,135].

### 3.10. Extracorporeal Shock Wave (ESW) Model

The integration of an advanced non-invasive approach, which involves an extracorporeal shock wave (ESW) and microbubbles, enables the induction of local and temporary TBI in discrete areas [136]. Additionally, ESW can induce thermal injury during shock wave treatment, but the conversion depth at higher frequencies is limited [137]. Shock waves initiate cavitation [138,139,140], which is considered to be the main mechanism involved in the local TBI during modeling [141,142], and the imitation of this mechanism does not require craniectomy [143]. Suitable shock wave devices are commercially available, which eliminates the need for developing complex ESW devices for modeling TBI [144]. To minimize the impact of the human factor, LabVIEW 2022is used, ensuring a coordinate accuracy of 0.1 mm (2.5 mm ± 0.1 mm) at various distances from the scalp. The structural arrangement of four positioning lasers on the outer perimeter of the device optimizes the positioning mechanism and improves accuracy [143,144].

A concave ESW probe is combined with gel buffer to ensure the localization of the shock wave focus at 5 mm from the proximal surface of the gel buffer. The probe combined with the gel buffer is positioned on the dorsal parietal surface of the rat’s head, providing the focus at 5 mm below the parietal surface and 3 mm caudal, lateral, or medial to the bregma suture of the rat’s skull (Table 3). Acoustic gel is applied to the interface between the probe and the gel buffer, and between the bottom of the gel pad and the rat’s scalp [93,144,145].

The ESW model is a non-interventional preclinical platform for modeling focal TBI, which is characterized by high reproducibility, rapidity, easy to use, and replication of injuries histologically identified as traumatic contusion and intracranial hemorrhage. Parameters and focusing adjustments help vary the intensity and location of injury, providing highly predictable results.

### 3.11. Blood Biomarkers in Animal TBI Models

Animal models provide the most uniform and reproducible method for studying TBI because experimental TBI models are free of confounding factors, such as age, medications, comorbidities, and polytrauma. Thus, changes in serum or plasma levels of biochemical markers can only be explained by injury if appropriate sham controls are used [66,146].

There are five neuroglial proteins used as biomarkers of tissue damage in the nervous system: neurofilament heavy polypeptide (NF-H), the glial fibrillary acidic protein (GFAP), ubiquitin carboxy-terminal hydrolase L1 (UCHL1), neuron-specific enolase (NSE), the myelin basic protein (MBP), the tau protein, and the S100β protein. These markers are elevated in the acute phase of severe head injury and are informative for prognosis [147,148,149,150,151].

In a porcine TBI model, changes in serum neurofilament heavy chain (NF-H) protein concentrations predicted injury severity and outcome. The study examined the time profiles of four serum protein biomarkers: S100β, NSE, the myelin basic protein (MBP), and NF-H over two weeks after injury. However, only a relative NF-H concentration correlated with the outcome. Compared with animals with favorable and unfavorable outcomes, serum NF-H concentrations peaked 6 h after injury, returned to baseline levels after three days, and remained at this level until the end of the two-week observation period [152].

Recent studies have attempted to extrapolate the correlation between plasma NF-H activity in an experimental TBI model, focusing on the hyperactive axonal form of NF-H (pNF-H). However, the obtained data did not reveal significant differences in the NF-H concentration in the adult population outside the acute injury period [153].

Experimental models of blast-induced TBI in rats and mice have demonstrated a correlation between the dynamics of serum biomarker concentrations and changes in metabolism, cell adhesion, extracellular matrix, proliferation, neuronal and glial damage, axonal degeneration, and inflammation [97]. Several specific biomarkers demonstrated complex and dynamically changing temporal patterns within a 30-day period after injury. These studies, along with others, have highlighted the importance of longitudinal biofluid profiling for monitoring the dynamics of biofluid biomarker concentrations following a single traumatic exposure. Therefore, single biomarker measurements may be misleading. Without serial sampling, key pathobiological processes, indicators of disease progression, and optimal therapeutic intervals may be missed [147,148,149,150].

## 4. Future Prospects for TBI Modeling

### 4.1. Modeling TBI in Aged Laboratory Animals

Historically, a disproportionate share of TBI-related injuries occurs in older adults. The hospitalization and death rates of people over 75 years of age exceed those of any other age group. However, despite the high prevalence of TBI in older adults, preclinical studies are predominantly conducted in relatively young animals. Obviously, age is a modifier of the outcome after TBI, and the development of aged animal models to study TBI is a promising area for further research [154,155].

In their article “Models of Traumatic Brain Injury in Aged Animals: A Clinical Perspective” (2019), Iboaya et al. provide the following relevant conclusions and results of a review of studies in laboratory mice and rats [156]:The mean lifespan of rats and mice in studies ranged from 24 to 30 months. Thus, extrapolating human age as a fraction of the total lifespan, a 20-month-old mouse or rat is equivalent to a 50–60-year-old human. The use of animals in the “old” age will allow covering the heterogeneity of the geriatric population, which is a specialized branch of gerontology devoted to the examination, prevention, and treatment of diseases of the elderly population.Modeling TBI should replicate the special mechanical properties typical of injuries in older adults. This should include studies that emulate concussion and mild to moderate TBI, which are the most common forms of clinical injury.Modeling TBI in combination with comorbidities frequently seen in older adults (e.g., hypertension, diabetes, cardiovascular disease). These comorbidities may be present in inbred and transgenic animals or induced in the laboratory settings (e.g., obesity, inactivity). Modeling of multiple comorbidities should also be considered.Modeling TBI in combination with medications commonly prescribed to older adults (e.g., preinjury antiplatelet therapy).Development of more accurate age-specific measures of functional outcomes in aged animals.Response to TBI should be assessed from acute to chronic post-TBI period.Use of biomarkers as outcome indicators in aged animals: measurements to confirm cerebrovascular reactivity and brain metabolism (magnetic resonance spectroscopy).Inclusion of both sexes in aged animal studies of TBI (rather than predominant use of males).Larger animal models of TBI should also be considered, including animals with gyrencephalic brains, such as sheep, pigs, and primates.

### 4.2. The Problem of Modeling TBI in Laboratory Animals

Modeling diffuse TBI in animals is limited by the induction of pathophysiological processes seen in humans due to the differences in size and anatomy between species. To extrapolate the model results to diffuse axonal brain injury in humans, injury parameters should be scaled depending on the animal size to replicate the mechanical impact characteristic of human TBI. The fact is that the larger mass of the human brain sustains more severe deformation and damage during acceleration than the smaller animal brain under the same acceleration/deceleration forces. For example, a scaling factor of up to 500% and 630% should be used for the baboon (140 g) and pig (90 g) brain, respectively, to model severe acceleration-induced diffuse axonal injury of the human brain. This calls into question the validity of head rotational acceleration models in small animals, in particular rodents. In rats (brain mass < 2 g), unattainable inertial forces (~8000% acceleration) are required to induce equivalent injury. However, there are small animal models of axonal injury that demonstrate clinically and morphologically relevant changes.

Animal TBI models have limitations, in particular the use of anesthesia which complicates the interpretation of traumatic pathology. In addition, certain human clinical manifestations following TBI, such as prolonged coma and post-traumatic depression, are not typically reproduced in rodent TBI models. However, despite these limitations, rodent TBI models remain valuable tools for research and development of new TBI treatments.

## 5. Conclusions

TBI causes significant morbidity and mortality. Animal TBI models provide a biological substrate for studying pathophysiology and consequences. They can be used to control biomechanical parameters and perform post-traumatic analysis. Animal TBI models are also valuable for studying disease-modifying therapeutics. The lack of specific treatment for TBI highlights the need for improved models that are as similar as possible in the course of pathophysiological processes [87]. The most recent models, e.g., CHIMERA, meet this need. The combination of TBI biomarker studies and animal TBI models will provide an important basis for the development of new diagnostic and treatment methods. The generalized rodent TBI models in this systematic review indicate tests that can be used in future studies to correlate pathology and behavioral changes in experimental animals.

## Figures and Tables

**Figure 1 biology-13-00813-f001:**
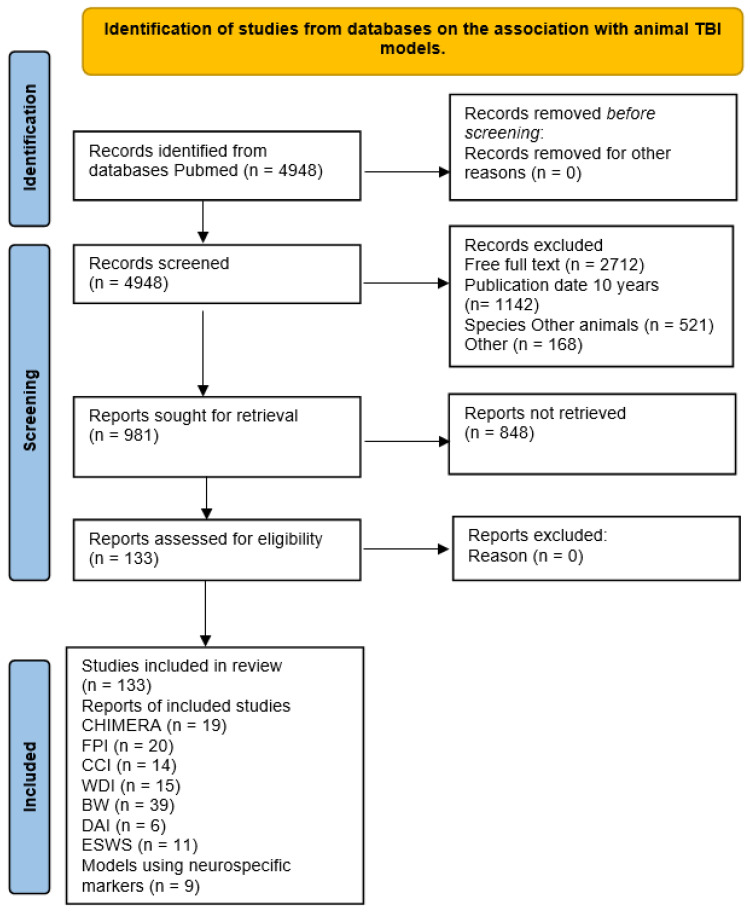
PRISMA flow diagram for the review.

**Table 1 biology-13-00813-t001:** Methods for inducing TBI in animal models.

Method	Advantages	Disadvantages	Key Findings	References
FPI	A good model for assessing focal, diffuse, or mixed, focal and diffuse, TBI	Experimental reproducibility depends on the accuracy of craniectomy	A limitation of animal TBI models on the basis of CCI, WD, and FPI methods is that injuries are usually induced by direct impact to the brain through a craniectomy, with the animal’s head being fixed. In this case, these conditions are not typical of human TBI. The size and location of a craniectomy are significant factors that can dramatically change the severity of injury, even when the same device is used to remove the aponeurosis.	[13,37,39,40,44,45,46,47,48]
	Possibility to assess behavioral outcomes in an animal model	Complex procedure		
	Evaluation of the efficiency of therapeutic remedies for translation into clinical practice	Development of pathological processes not associated with TBI		
		High mortality due to brainstem damage		
CCI	An animal focal injury model is the method of choice for studying TBI caused by direct blunt impact	A 5 mm trephination (injury window) is required		[13,49,50,51,52]
	Control and uniformity of injury	Hemorrhage and ischemia		
	No risk of re-injury	Blood–brain barrier disruption caused by tissue destruction		
	Damages persist up to 1 year due to brain atrophy and a progressive decrease in cerebral blood flow			
		Development of an acute or chronic neurodegenerative condition		
	Study reproducibility	Cognitive decline		
WDI	Severity of TBI is controlled via the height and weight of the load	Unintentional skull fracture		[13,46,48,53,54,55,56,57,58]
		Risk of a second rebound injury		
	A cost-effective and relatively easy to use model	Increased mortality rate		
		Lack of repeatability of the animal model		

**Table 2 biology-13-00813-t002:** Methods for modeling blast wave impact on animals.

Method	Advantages	Disadvantages	Key Findings	References
Shock tube	Generated energy does not dissipate	Injury in rodents should be scaled to comparable injury in humans	Differences in rodent models and human injury become noticeable when modeling repeated mild TBI, which requires accounting for differences in temporal pathology. In particular, an increase in vulnerability in rodents occurs at the hourly scale, whereas that in humans occurs at the daily scale. Additionally, injury classification in animal TBI models (mild, moderate, and severe) is not standardized.	[98,101,102,103,104,105,106]
	High repeatability of the animal model	There is no possibility to create peak pressure of the same intensity as that from a blast in an open field		
	There are no secondary or tertiary effects of blast injury	Differences between animals and humans in the brain surface, mass, geometry, white/gray matter ratio, and size		
	Provides precise control of blast wave intensity	No opportunity to generate a polytrauma model		
		No standards of explosivs, tube design, species, location in the tube, body shielding, and head mobility		
Open-field blast (BI)	Impulse shock wave (DAS)	Dissipation of generated energy		[66,107,108,109,110,111,112,113]
	Exponential pressure reduction (MPA method)	Blast source is quickly weakened		
	Possibility of a polytrauma model	Prolonged preparation for the experiment		
	Friedlander signal generation and MPA method	No standards of explosives		
	Spherical distribution of the blast wave in three-dimensional space	Differences between animals and humans in the brain surface, mass, geometry, white/gray matter ratio, and their sizes		

**Table 3 biology-13-00813-t003:** Main parameters of shock wave intensity levels for TBI modeling.

Intensity Level	8	10	16	20
Negative peak pressure (MPa)	−10.92	−12.9	−14.21	−18.7
Positive peak pressure (MPa)	23.1	28.5	48.1	77.7
Energy flux density (total) (MJ/mm^2^)	0.27	0.35	0.6	0.82

## Data Availability

Not applicable.

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
