# Peer review of "A Systematic Review of Traumatic Brain Injury in Modern Rodent Models: Current Status and Future Prospects"

_biology, 2024, doi:10.3390/biology13100813_

Round 1

Reviewer 1 Report

Comments and Suggestions for Authors

This is an interesting study which adheres to ethical standards.

The manuscript is devoted to the advantages and disadvantages of modern rodent models of traumatic brain injury. Although the review is entitled A Systematic Review of Animal Tbi Models: Current Status and Future, it focuses only on rodent models. Since there are other animal models (for instance, in zebrafish), the title should be corrected and focus on rodent models instead of other animal models should be discussed in more details.
Methods of query in databases (PubMed and WOS) and selection criteria are described in sufficient details; the authors provide a flow-chart of the selecting of the paper analysis according to the PRISMA guidelines. The review gives a good overview of the current state of art in modelling TBI in rodents. The authors analysed the key findings in different models, their advantages and disadvantages. The review would be of interest and helpful for neuroscientists who study TBI, preclinical and translational research in the field.

Author Response

Comments: The manuscript is devoted to the advantages and disadvantages of modern rodent models of traumatic brain injury. Although the review is entitled A Systematic Review of Animal Tbi Models: Current Status and Future, it focuses only on rodent models. Since there are other animal models (for instance, in zebrafish), the title should be corrected and focus on rodent models instead of other animal models should be discussed in more details.

Methods of query in databases (PubMed and WOS) and selection criteria are described in sufficient details; the authors provide a flow-chart of the selecting of the paper analysis according to the PRISMA guidelines. The review gives a good overview of the current state of art in modelling TBI in rodents. The authors analysed the key findings in different models, their advantages and disadvantages. The review would be of interest and helpful for neuroscientists who study TBI, preclinical and translational research in the field.

Response We sincerely appreciate your careful review of our work and the valuable comments provided to enhance the quality of the manuscript. 

We have revised the title of the article to "A Systematic Review of TBI in Modern Rodent Models: Current Status and Future Prospects." The updated title more precisely aligns with the content of the study in its current version.

Reviewer 2 Report

Comments and Suggestions for Authors

Dear sir,

I have the following comments regarding your article 

Introduction:

1. Page 1, Line 33 - Traumatic brain injury may be mentioned in its entirety along with the short form TBI, the first time it is mentioned. Subsequently, only short form TBI can be used.

Materials and methods:

1. Page 3, Figure 1: Please provide a clear picture with the text clearly visible in the picture. Many words are unreadable. Specially, please correct the text in the “preliminary ghjvcjnh (N=654)” box.

2. Methodology could be further explained in a more detailed way.

3. There is no clarity about how the 167 articles were summarized into 133 articles. Please explain so that the readers will not be having any confusion. (Page 4, Lines 113-115)

Results:

I. TBI Pathogenesis

a. Page 4, Lines 133-136 describe what is known as Monro-Kellie doctrine. Please mention it.

II. TBI Models

a. Page 5, Lines 184-185: Please correct the sentence “In addition, a short lifespan of 2 to 3 years in most mouse and rat strains provides a significant advantage for aging research[13] and aspects will be discussed in more detail below.” With respect to grammatical errors.

III. Fluid percussion brain injury (FPI) model

a. Page 7, Line 212: Probably the word is written in some other language. Please correct the word “реакций”

IV. Blast wave (BW) model of TBI

a. Page 8, Line 291 – “one of the most” not “ones of the most”

b. Page 8, Line 296 – “Of these, widely popular laboratory are those....” Please correct the sentence for grammatical error. Probably there is a word missing after laboratory.

c. Page 11, Line 356 – “... impact ones are the most important” not “... impact ones are most important”

Author Response

We sincerely appreciate your thorough review of our work and the constructive comments provided to enhance the quality of the manuscript. Your feedback has been invaluable in helping us refine and improve the paper, and we have carefully considered each suggestion. We believe that the revisions made in response to your recommendations have strengthened the clarity and depth of the study. Thank you again for your time and effort in reviewing our manuscript. We hope that the updated version now meets your expectations

Comments 1.

Introduction:

  1. Page 1, Line 33 - Traumatic brain injury may be mentioned in its entirety along with the short form TBI, the first time it is mentioned. Subsequently, only short form TBI can be used.

Response 1:  Thank you for your valuable remarks. We have implemented the correction by fully spelling out "traumatic brain injury" (TBI) along with its abbreviated form.

Comments 2.

Materials and methods:

  1. Page 3, Figure 1: Please provide a clear picture with the text clearly visible in the picture. Many words are unreadable. Specially, please correct the text in the “preliminary ghjvcjnh (N=654)” box.

Response 2.1:  We appreciate your remarks. In response, we have updated Figure 1 in line with the PRISMA flowchart guidelines and made the corresponding revisions to the main text.

  1. Methodology could be further explained in a more detailed way.

Response 2.2:  We value your recommendation and have accordingly made the necessary adjustments to the "Materials and Methods" section.

  1. There is no clarity about how the 167 articles were summarized into 133 articles. Please explain so that the readers will not be having any confusion. (Page 4, Lines 113-115):

Response 2.3:   We appreciate your comment. In response, we have refined the "Results" section to enhance clarity and ensure better understanding for the readers.

Comments 3.

Results:

I. TBI Pathogenesis

a. Page 4, Lines 133-136 describe what is known as Monro-Kellie doctrine. Please mention it.

Response 3.I. We appreciate your important input. We have incorporated the necessary information into the relevant section, where we have addressed the Monro-Kellie doctrine.

II. TBI Models

a. Page 5, Lines 184-185: Please correct the sentence “In addition, a short lifespan of 2 to 3 years in most mouse and rat strains provides a significant advantage for aging research[13] and aspects will be discussed in more detail below.” With respect to grammatical errors.

Response 3.II. We appreciate your comment. The appropriate revisions have been made to the manuscript.

III. Fluid percussion brain injury (FPI) model

  1. Page 7, Line 212: Probably the word is written in some other language. Please correct the word “реакций”

Response 3.III Thank you for your valuable remarks. The duplicated word in the foreign language has been removed.

IV. Blast wave (BW) model of TBI

a. Page 8, Line 291 – “one of the most” not “ones of the most”

Response 3.IV.a Thank you for your observation. We have made the necessary correction to the manuscript.

b. Page 8, Line 296 – “Of these, widely popular laboratory are those....” Please correct the sentence for grammatical error. Probably there is a word missing after laboratory.

Response 3.IV.b. Thank you for your observation. We have implemented the necessary corrections in the manuscript.

c. Page 11, Line 356 – “... impact ones are the most important” not “... impact ones are most important”

Response 3.IV.c. Thank you for your comment. We have revised the text and made the necessary corrections.